# CPAP Treatment Improves Pure Tone Audiometry Threshold in Sensorineural Hearing Loss Patients with Sleep-Disordered Breathing

**DOI:** 10.3390/ijerph18136768

**Published:** 2021-06-24

**Authors:** Jessie Chao-Yun Chi, Shin-Da Lee, Ren-Jing Huang, Ching-Hsiang Lai, Stanley Yung Liu, Yih-Jeng Tsai, Po-Han Fu, Hua Ting

**Affiliations:** 1Institute of Medicine, Chung Shan Medical University, Taichung 40201, Taiwan; hawk.shaw@msa.hinet.net; 2Department of Otorhinolaryngology, Head and Neck Surgery, Taichung Hospital, Ministry of Health and Welfare, Taichung 403, Taiwan; 3Department of Physical Therapy, Graduate Institute of Rehabilitation Science, China Medical University, Taichung 40402, Taiwan; shinda@mail.cmu.edu.tw; 4Department of Occupational Therapy, Asia University, Taichung 413, Taiwan; 5School of Rehabilitation Medicine, Shanghai University of Traditional Chinese Medicine, Shanghai 200032, China; 6Department of Medical Image and Radiological Sciences, Chung Shan Medical University, Taichung 40201, Taiwan; huangrenjing@yahoo.com.tw; 7Department of Medical Informatics, Chung Shan Medical University, Taichung 40201, Taiwan; 8Department of Otolaryngology, Stanford University School of Medicine, Stanford, CA 94305-5101, USA; ycliu@stanford.edu; 9School of Medicine, Fu-Jen Catholic University, New Taipei City 24205, Taiwan; tigeryjtsai@gmail.com; 10Department of Otolaryngology Head and Neck Surgery, Shin Kong Wu Ho-Su Memorial Hospital, Taipei 111045, Taiwan; 11Graduate Institute of Photonics and Optoelectronics, National Taiwan University, Taipei 10617, Taiwan; pohan521@gmail.com; 12Department of Physical Medicine and Rehabilitation, Chung Shan Medical University Hospital, Taichung 40201, Taiwan; 13Sleep Medicine Center, Chung Shan Medical University Hospital, Taichung 40201, Taiwan

**Keywords:** continuous positive airway pressure, obstructive sleep apnea, neural deafness, pure-tone audiometry, polysomnography

## Abstract

This article investigates the effects of continuous positive airway pressure (CPAP) on hearing impairment in sensorineural hearing loss (SNHL) patients with sleep-disordered breathing (SDB). This retrospective and observational study took place from September 2016 to February 2021, accumulating 77 subjects with SNHL and SDB (60.7 ± 11.1 years). Of which, 28 received CPAP treatment (63.0 ± 8.5 years). In our methodology, hearing thresholds at low, medium, high, and average frequencies are assessed by pure-tone audiometry at baseline (BL), three (3 m), six (6 m), and 12 (12 m) months. Our results show that the BL of at least three frequencies in all subjects is positively associated with old age, males, smoking, alcohol, coronary artery disease, hypertension, and apnea-hypopnea index [AHI] (all *p* < 0.05). Moreover, low, medium, and average frequencies are negatively correlated at CPAP-6 m (−5.60 ± 2.33, −5.82 ± 2.56, and −5.10 ± 2.26 dB; all *p* < 0.05) and CPAP-12 m (−7.97 ± 2.74, −8.15 ± 2.35, and −6.67 ± 2.37 dB; all *p* < 0.01) against corresponding measures of CPAP-BL. High, medium, and average frequencies positively correlated with age (*p* < 0.001 for high and average frequencies and <0.01 for medium frequencies). We conclude that in SNHL patients with SDB, hearing thresholds at low and medium frequencies improves under CPAP use after six months, which persists at least to the end of one year.

## 1. Introduction

Sensorineural hearing loss (SNHL), resulting from cochlear hair cell loss and associated auditory nerve dysfunction [1], is one of the most common geriatric disorders. Hearing loss resulting in impaired verbal communication could be linked to social isolation, anxiety, depression, cognitive impairment, and reduced quality of life [2,3]. SNHL has been cited [4] in recent years as the third leading cause of disability worldwide by the World Health Organization. SNHL is associated with genetic [5], vascular, hormonal, toxic, neoplastic, infectious, traumatic, and iatrogenic factors, as well as smoking and alcohol habits, systemic diseases, environmental noise, and socioeconomic status [6,7,8]. SNHL patients present with degenerative changes (at least partially caused by aging), earlier in the central than the peripheral auditory system [9,10]. It is crucial to find a way to slow or prevent the worsening of hearing impairment in geriatric individuals to maintain their quality of life.

Sleep-disordered breathing (SDB) is characterized by repetitive complete or partial upper airway obstruction during sleep, possibly together with symptoms of excessive daytime somnolence, decreased alertness, memory deficits, and depression [11]. It affects 24% of middle-aged males and 9% of middle-aged females [12], and is more prevalent in the elderly. The repeated episodes of apnea and hypopnea may trigger intermittent hypoxia, promote reactive oxygen species production and the inflammatory cascade, and advance vascular endothelial injury. SDB is also associated with hypertension, cardiovascular diseases, neurocognitive disorder, peripheral neuropathy [13], and even ocular neuropathy [14]. Therefore, growing evidence has unsurprisingly supported the relationship between SDB and hearing impairment [15,16,17,18]. SDB specifically, rather than simply snoring [19,20], has been independently associated with hearing impairment [17], sudden SNHL [21], and central auditory dysfunction in the elderly [22]. Continuous positive airway pressure (CPAP), the primary therapy for SDB, has been shown to prevent intermittent hypoxemia, recover sleep quality, and improve health-related quality of life [23]. Intriguingly, uvulopalatopharyngoplasty may reverse [24] the low prevalence of transient-evoked otoacoustic emissions and reduced distortion product otoacoustic emissions, which both are associated with SDB. When CPAP was used for six months, it was shown to mitigate reduced hearing ability from Meniere disease [25]. Based on pathophysiological rationales, CPAP therapy for a specific period of time should potentially have a beneficial effect on the hearing impairment of patients with SNHL and SDB. Nevertheless, the effect of CPAP therapy on hearing impairment in SNHL patients has not been investigated.

We hypothesized that for patients with subacute or chronic hearing impairment and coexistent SDB, CPAP therapy would improve the pure tone audiometry threshold, regardless of age and SDB severity.

## 2. Materials and Methods

### 2.1. Study Design

This retrospective and observational study was conducted at a single hospital from September 2016 to February 2021. The study was approved by the Institutional Review Board of the Tsao-tun Psychiatric Center, Taichung, Taiwan (No. 108002).

### 2.2. Study Population

One hundred forty-eight consecutive patients applying for social subsidies and/or clinical services with chronic hearing impairment or sudden hearing loss were recruited from our Otorhinolaryngology clinics (Figure 1).

The inclusion criteria were SNHL patients of both gender who were over 18 years old, had symptoms of SDB, and were willing to participate in our study. SNHL was defined as hearing thresholds ≥ 25 decibels (dB) uni- or bilaterally at any tested frequency by pure-tone audiometry (PTA). Air conduction and bone conduction were calculated, and conductive hearing impairment was excluded from our study. Symptoms of SDB [11,26] include loud snoring, excessive daytime sleepiness, nocturia, sleep interruption, and mouth breathing when sleeping.

The exclusion criteria were (1) a history of any condition potentially leading to conductive hearing loss [27] (external or middle ear infectious disease, eardrum perforation, otosclerosis, external or middle ear tumor, or ossicular chain dislocation); the air-bone gap was also calculated to exclude conductive hearing impairment precisely. (2) Hearing loss induced by ototoxic drugs or sudden deafness under high dose steroid treatment or *hyperbaric oxygen* (HBO) therapy. We excluded those etiologies of SNHL which might be affected by the medications [27,28], such as steroid, aminoglycosides, cisplatin, furosemide, and so on. (3) Disorders or administration of medications or substances that could affect neurocognitive function; or (4) current or prior SDB treatment with CPAP or surgery.

### 2.3. Protocol

Information related to anthropometric measurements, seated blood pressure, current medications, general health data, hearing-related history, sleep symptoms, as well as otorhinolaryngological examinations and services, time points for otoscopy, and audiometry testing, and the night of the polysomnography (PSG) study were obtained on the chart. Relevant covariates [5,29] of interest for SNHL and SDB were also recorded: age, gender, body mass index, cigarette smoking, alcohol consumption, coronary artery disease, hypertension, diabetes mellitus, hypercholesterolemia, depression, sudden deafness, and the duration of hearing impairment. PTA was conducted in the otorhinolaryngological lab within three days after the first clinical visit (baseline measure; BL). The PSG study was performed in the sleep lab within 10 days for those who fulfilled the inclusion criteria, and had no exclusion criteria.

All eligible subjects were further separated into two groups, CPAP and non-CPAP treatment. The former included those with good primary adherence to CPAP and regular medications for personal, systemic illnesses based on the chart record. However, regardless of receiving the CPAP treatment or not, all the patients received sleep hygiene education and adequate medications [30], such as nasal spray for nasal obstruction before sleep and regular regimens for individual systemic diseases. In all participants, PTA hearing thresholds were re-evaluated at three (3 m), six (6 m), and 12 (12 m) months. CPAP compliance was defined as using the therapy for at least an average of four hours per night, as determined by the card reader.

### 2.4. Pure-Tone Audiometry (PTA)

PTA was performed to determine individual hearing thresholds using an annually calibrated GSI 61 audiometer (Grason-Stadler, Eden Prairie, MN 55344, USA) equipped with earphones (Telephonics, Farmingdale, New York, NY, USA). Hearing thresholds were tested at six pure-tone frequencies (250, 500, 1000, 2000, 4000, and 8000 Hz for air conduction). A qualified audiologist blinded to the treatment assignments performed all the auditory testing procedures in a double-walled, soundproof booth. An ear specialist (Dr. Chi) supervised all data collection and interpretation of audiometry studies.

To quantify the severity of hearing impairment, the hearing threshold frequencies of interest were categorized as low frequency threshold (low; 250 Hz), medium frequency (medium; mean of 500, 1000, and 2000 Hz), high frequency (high; mean of 4000 and 8000 Hz), and the mean of these six frequencies (average). To avoid bias, hearing threshold measurements were always taken from the same ear, selected by the likelihood of responding to treatment. If the patient had sudden unilateral SNHL, we chose that affected ear after two weeks of standard steroid treatment.

To further distinguish the effect of time, all subjects received three follow-up pure-tone audiometry assessments at three, six, and 12 months (3 m, 6 m, and 12 m) in addition to a baseline (BL).

### 2.5. Polysomnography (PSG)

Participants were asked to abstain from caffeinated food and drinks after lunchtime on the day of sleep testing as our routine. They arrived at the sleep center at 9:00 p.m. to complete anthropometric assessments, questionnaires for chronic psychophysiological illnesses, and physical exams. Sleep was evaluated by PSG between 10:30 p.m. and 6:00 a.m. As in our previous studies [31], the procedures, skilled professionals, electroencephalography dependent sleep staging, and scoring criteria of the sleep variables were defined according to the 2007 American Academy of Sleep Medicine Manual guidelines. The sleep variables [32] consisted of sleep efficiency, total sleep time, apnea-hypopnea index (AHI), arousal index, lowest oxygen saturation (miniSpO_2_), percentage of the total period for which oxygen saturation was less than 90%, rapid eye movement (REM), non-REM (stage 1, 2, and slow-wave sleep). Hypopnea was defined as a 30 to 90% reduction in oronasal flow for at least 10 s, followed by at least a 3% decrease in arterial oxygen saturation. A sleep study was considered effective if there was more than four hours of recording and more than three hours of total sleep time recorded.

### 2.6. Continuous Positive Airway Pressure (CPAP)

The patients tried CPAP therapy at home with either the Auto A-Flex (Philips Respironics System One; “AutoSet”, Murrysville, PA, USA) or the SleepStyle Auto (Fisher and Paykel Healthcare SleepStyle, Auckland, New Zealand) machines with a pressure range of 4 to 20 cm H_2_O. For automatic pressure titration during initiative therapy in the first week, the pressure was set at 4 to 15 cm H_2_O and then adjusted to an individualized optimal constant pressure based on a review of CPAP use. All subjects were provided humidifiers to avoid upper airway dryness. A nasal or full-face mask was used based on user preference, final titrated pressure, and bed partner’s observation. Site staff contacted users three times within the first week after starting CPAP to encourage usage adherence and manage relevant problems.

### 2.7. Illnesses and Their Definitions

The covariates for hearing impairment [5,6,7,8,9,10] of interest consisted of age, gender, obesity, and the presence of various chronic psychophysiological illnesses. Coronary artery disease and depression were defined using self-reported history and current regular medicines. Hypertension was defined [33] as having a systolic or diastolic blood pressure ≥ 140/90 mmHg or current use of antihypertensive medications. Diabetes mellitus was defined [34,35] as having a fasting glucose ≥ 126 mg/dL, non-fasting glucose ≥ 200 mg/dL, hemoglobin A1C ≥ 6.5%, or current diabetes medication use. Hypercholesterolemia [36] was defined as having a low-density lipoprotein cholesterol ≥ 160 mg/dL, high-density lipoprotein cholesterol ≤ 40 mg/dL, triglycerides ≥ 200 mg/dL, or current relevant medication use.

### 2.8. Statistical Analyses

All continuous data were presented as mean ± standard deviation. The Student’s *t*-test and chi-squared test were used for continuous and categorical data, respectively, to compare the differences in various parameters between the non-CPAPand CPAP groups. Univariate analyses to examine relationships between continuous or categorical variables and baseline hearing thresholds of various frequencies were conducted using Pearson’s correlation and one-way analysis of variance (ANOVA) [37], respectively.

The traditional statistics and a generalized estimating equation (GEE) [38,39] were used to examine changes or relationships between hearing thresholds in the pure tone audiometry at various frequencies and the participants’ data in the CPAP group with adjusting the covariate factors.

Statistical analysis was performed using SPSS 25.0 (Armonk, NY, USA: IBM Corp). Two-sided *p* values < 0.05 were considered statistically significant.

## 3. Results

One hundred forty-eight consecutive patients with SNHL and SDB symptoms were initially selected, 71 patients were excluded because of incomplete chart data (Figure 1). Of these, 48 individuals were excluded, due to loss follow-up and an incomplete PTA or PSG examination, 19 were not SDB sufferers, and four had prior upper airway surgeries. Seventy-seven subjects were recruited to our observational study.

Of these 77 subjects (Table 1; 54.5% male, 60.7 ± 11.1 years, 25.2 ± 4.2 kg/m^2^, 28.0 ± 22.8 events/hour in AHI, 82.5 ± 8.7% in miniSpO_2_, 32.9 ± 19.6, 32.3 ± 22.4, 54.4 ± 24.6 and 40.2 ± 21.3 dB in BL hearing thresholds at low, medium, high, and average frequencies, respectively), 28 received CPAP therapy with good compliance throughout the study. The remaining participants were in the non-CPAP treatment group (N = 49). However, because the patients could decide to receive CPAP treatment or not by themselves, most of the patients who had mild to moderate SDB didn’t want to receive CPAP treatment. That’s why there was a significant difference in AHI between these two groups. Additionally, there was a significant difference in BMI, miniSpO2, gender, cigarette smoking, alcoholic drinking, and hypertension in these two groups. Those factors were correlated with the severity of sleep apnea. Because of the severity of SDB and some parameters not being similar, it was not possible to compare them as intervention and control groups. Therefore, to discuss the correlation between PTA threshold and CPAP treatment in SNHL patients, we discussed CPAP groups (N = 28) only. There was attrition of participants in 6 M and 12 M because of poor CPAP compliance. To be more specific, there were still 28 participates in three months. In six months, one participate was excluded because of poor CPAP compliance (N = 27). In twelve months, four participates were excluded, due to the same reason (N = 24).

To discuss the relationship between covariate factors and hearing thresholds, among 77 participants, the covariate factors of SNHL (Old age, male, smoking, alcohol, coronary artery disease, hypertension, and AHI; all *p* < 0.05), were positively associated with baseline hearing thresholds at more than three frequencies among all subjects (Table 2). Interestingly, high, medium, and average frequencies positively correlated with age (*p* < 0.001 for high and average frequencies and <0.01 for medium frequencies). Concerning AHI, it was related to low, medium, high, and average frequencies all. Remarkably, depression seemed to be related to thresholds at medium frequency. However, the sample size was too small (N = 5) to conduct further statistical analyses. In Table 2, the mean ± SD was shown in the nominal scale, and Pearson product-moment correlation coefficient (r) was calculated in the ratio scale.

To reiterate, to avoid potential bias in the CPAP and non-CPAP groups with significant differences, the PTA thresholds were analyzed in the CPAP group (N = 28) only at low, medium, high, and average frequencies in the baseline, 3 months, 6 months, and 12 months after CPAP treatment. The traditional statistics and a generalized estimating equation (GEE) were used to examine changes or relationships between hearing thresholds in the pure tone audiometry at various frequencies and the participants’ data in the CPAP group with adjusting the covariate factors. (Table 3 and Figure 2). The covariate factors, which were showed in Table 2, such as age, gender, smoking, alcohol, coronary artery disease, hypertension, and AHI, were adjusted. Low, medium, and average frequencies were negatively associated at CPAP-6 m (−5.60 ± 2.33, −5.82 ± 2.56, and −5.10 ± 2.26 dB, respectively; all *p* < 0.05) and CPAP-12 m (−7.97 ± 2.74, −8.15 ± 2.35, and −6.67 ± 2.37 dB, respectively; all *p* < 0.01) with corresponding measures of CPAP-BL).

## 4. Discussion

Our study shows that for SNHL patients with coexistent SDB, good compliance with CPAP treatment for 6 to 12 months may improve pure tone audiometry threshold at low, medium, and average frequencies, with adjusting age, gender, smoking, alcohol, coronary artery disease, hypertension, and AHI. While low, medium, high, and average frequencies were significantly correlated with AHI, which indicated the severity of SDB; high, medium, and average frequencies were significantly related to age. We postulate that treating SDB might reverse hearing deficit on the apex and middle turns of the cochlea, rather than the basal turn, which is mainly worsened by the aging process [40].

Indeed, SDB had been reported associated with hearing problems and possibly sudden deafness [41]. Although the lowest oxyhemoglobin saturation at night sleep possibly the factor is affecting hearing impairment [16], AHI exhibited dose-responsively associated with hearing impairment at various frequencies in a previous huge study (~14,000 participants) [17]. High frequency hearing loss was detected in adults with severe SDB with prolonged latencies of waves I and V of auditory brainstem evoked response [42]. Another longitudinal observational study [43] of 6797 individuals, showed that SDB increases the odds of a future hearing impairment by 21%. Some authors also noted that hearing threshold values in the patients with SDB were significantly higher compared to the control group at 500, 1000, 2000, 4000, and 8000 Hz in both ears [44]. The finding of the present study that AHI is the only factor beyond age-related to the hearing threshold after adjusting for all anthropometric factors, is quite consistent with previous cross-sectional studies [17,42]. However, it is still a question if treating SDB effectively could attenuate the hearing impairment in SNHL patients, or even improve the hearing threshold. The results of our study might offer the preliminary data in this issue, though no consistent effect was found in another small retrospective study (n = 12 and 10 in CPAP and non-CPAP groups, respectively) [45]. Nevertheless, future randomized control study is warranted.

A murine model of four-week long sleep apnea [46] leaded to extensive damage over the outer hair cells of the organ of Corti. The community study mentioned above [17] has shown that patients with moderate to severe SDB had hearing impairment, particularly at high frequency, regardless of age. The moderate level of SDB affected high frequency hearing abilities, whereas the severe level affected all hearing functions [18]. High frequency hearing loss in severe SDB was caused by cochlear damage [15]. Severe OSA is independently associated with cochlear function impairment in patients with no significant co-morbidities [47]. All these findings might support hair cells in the basal turn of the cochlea being more susceptible to SDB-associated oxidative stress [48], probably due to the lower activity of glutathione-related antioxidant enzymes [49]. In contrast, the present finding that high frequency hearing impairment is independently correlated with age [40,49]. We inferred that in addition to aging effects, the cochlear base in our patients was damaged irreversibly by SDB, while the cochlear apex and middle turns (corresponding to low frequency hearing) were relatively reversible after 6 to 12 month long CPAP therapy.

To date, there are no effective treatments for SNHL [50], with the exception of hearing aids, cochlear implants, and glucocorticoid or hyperbaric oxygen treatment for sudden SNHL. Insulin-like growth factor-1 (IGF-1) [50] recently emerged as a possible factor for treatment of SNHL because it depletes by the mutation of a corresponding human homozygous gene resulting SNHL, negatively correlated with SDB-related hypoxemia [51], and increased in secretion after three months and six months of CPAP use for SDB patients. In our study, the improvement by CPAP treatment of hearing at 6 m and 12 m may possibly reflect restorations of the IGF-1 axis, as well as the apical cochlear function. Undoubtedly, further relevant clinical studies are needed to prove.

Auditory evoked potentials would be affected at somewhat higher centers than at the cochlear level [52]. The prefrontal cortex [53] also appears vulnerable to sleep fragmentation and intermittent hypoxia caused by SDB. Apart from higher pure-tone thresholds, severe SDB subjects were likely to experience ponto-mesencephalopathy [54], higher speech recognition thresholds, and lower speech discrimination ability [18]. In fact, hearing thresholds, speech recognition thresholds, and speech discrimination disability were all positively correlated with AHI [22] and desaturation index [55], and negatively correlated with minimum oxygen saturation [18]. Further clinical and basic science studies focusing on the mechanism are warranted.

### Limitations

The single-hospital data collection of a relatively small number of Taiwanese subjects with a wide range of body mass index and AHI limits its generalizability. The significant difference in demographic measures between CPAP and non-CPAP groups could lead to a bias. Thereafter, the case series study was chosen for data processing. Further randomized control cohort studies of a larger number of participants were expected in the future. Additionally, the reliability of self-reported lifestyles and the presence of illnesses might bias the results.

Some strengths are also noteworthy: Sleep and hearing impairment measured by standardized protocols [56,57], accurate CPAP follow-up, findings adjusted for confounding variables, the duration of treatment, and follow-up longer than most previous studies.

## 5. Conclusions

In summary, CPAP application to SNHL patients with comorbid SDB improved hearing threshold at low, medium, and average frequencies after three months, reliably present at six months, and the effects persist throughout the first year. CPAP therapy is a potentially important treatment for individuals with SNHL and SDB. Further investigation is required to determine whether early application of CPAP alone or combined with topical administration of any neurotrophic factor, such as IGF-1, can serve as a valid strategy for preventive or therapeutic purposes.

## Figures and Tables

**Figure 1 ijerph-18-06768-f001:**
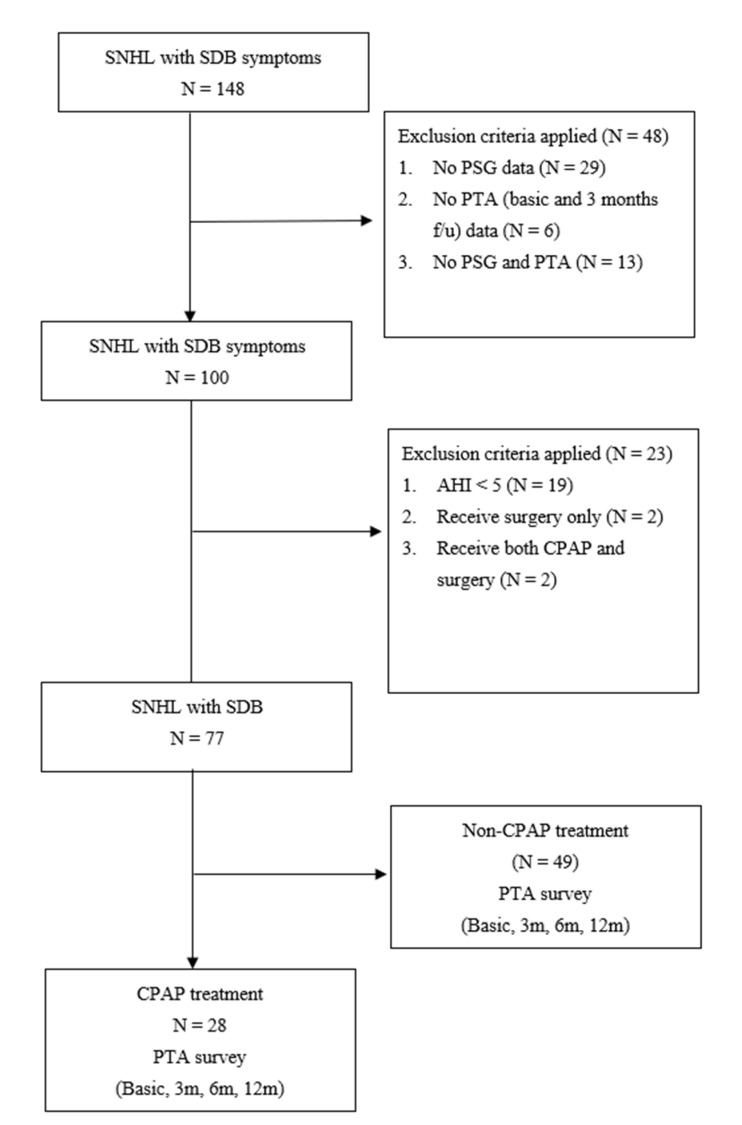
Flow chart showing recruitment methodology. Hearing thresholds were measured by pure-tone audiometry (PTA) at baseline (BL), three (3 m), six (6 m), and 12 months (12 m) in the 77 patients. *Abbreviations*: SNHL, sensorineural hearing loss; SDB, sleep-disordered breathing; PSG, sleep polysomnography study; CPAP continuous positive airway pressure therapy group.

**Figure 2 ijerph-18-06768-f002:**
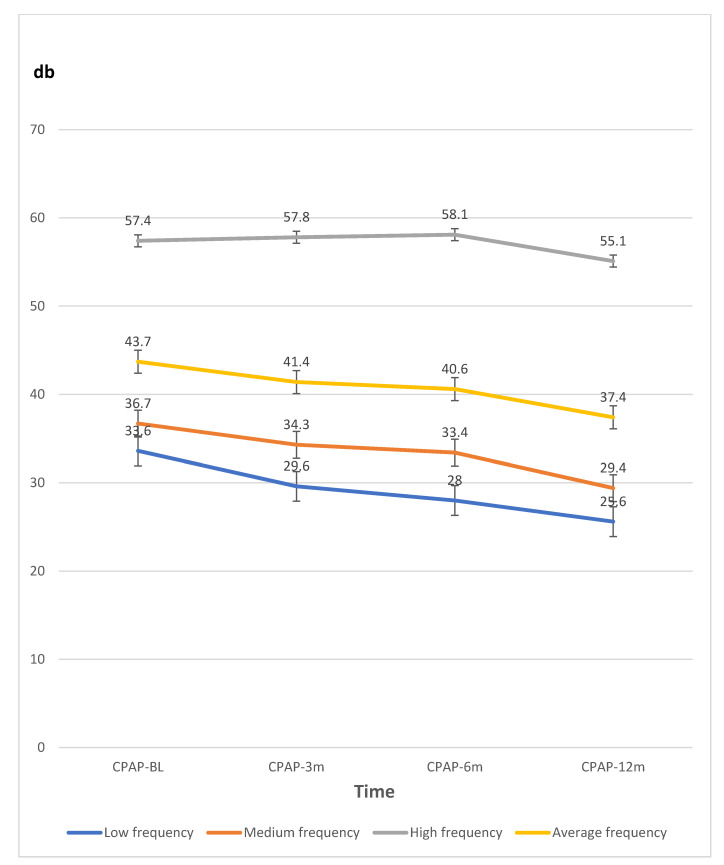
The line graph data of hearing thresholds by pure tone audiometry (dB) at various frequencies measured at baseline (BL), three (3 m), six (6 m), and 12 months (12 m) in the CPAP therapy group. *Abbreviations*: CPAP continuous positive airway pressure therapy; db, decibels.

**Table 1 ijerph-18-06768-t001:** Differences between non-CPAP group and CPAP group in anthropometric variables, comorbid diseases, history and severity of hearing impairment, and severity of sleep-disordered breathing.

	Total (N = 77)			Non-CPAP (N = 49)			CPAP (N = 28)			
	Mean ± SD; N (%)			Mean ± SD; N (%)			Mean ± SD; N (%)			*p*
Age, years	60.7	±	11.1	59.3	±	12.3	63.0	±	8.5	NS
Body mass index, kg/m^2^	25.2	±	4.2	24.2	±	3.9	26.8	±	4.3	**
Apnea-hypopnea index, events/hour	28.0	±	22.8	19.8	±	18.8	42.4	±	22.3	***
miniSpO_2_, %	82.5	±	8.7	84.1	±	8.2	79.7	±	9.0	*
Male, N (%)	42		(54.5)	22		(44.9)	20		(71.4)	*
Smoking, N (%)	20		(26.0)	9		(18.4)	11		(39.3)	*
Alcohol, N (%)	27		(35.1)	12		(24.5)	15		(53.6)	*
Coronary artery disease, N (%)	18		(23.4)	8		(16.3)	10		(35.7)	NS
Hypertension, N (%)	33		(42.9)	15		(30.6)	18		(64.3)	**
Diabetes mellitus, N (%)	12		(15.6)	7		(14.3)	5		(17.9)	NS
Hypercholesterolemia, N (%)	14		(18.2)	8		(16.3)	6		(21.4)	NS
Depression, N (%)	5		(6.5)	2		(4.1)	3		(10.7)	NS
Sudden deafness, N (%)	20		(26.0)	10		(20.4)	10		(35.7)	NS
Duration of hearing impairment										NS
<6 months, N (%)	29		(37.7)	12		(24.5)	7		(25.0)	
6 months~3 years, N (%)	18		(23.4)	5		(10.2)	13		(46.4)	
>3 years, N (%)	30		(39.0)	11		(22.4)	19		(67.9)	
Baseline measures of hearing threshold										
Low frequency, dB	32.9	±	19.6	32.6	±	19.1	33.6	±	20.7	NS
Median frequency, dB	32.3	±	22.4	29.8	±	21.2	36.7	±	24.1	NS
High frequency, dB	54.4	±	24.6	52.7	±	24.1	57.4	±	25.6	NS
Average frequency, dB	40.2	±	21.3	38.1	±	20.2	43.7	±	22.9	NS

Abbreviations: CPAP, continuous positive airway pressure therapy group; SD, standard deviation; miniSpO_2_, lowest oxyhemoglobin saturation; dB, decibels. * *p* < 0.05, ** *p* < 0.01, *** *p* < 0.001, and NS: non-significance.

**Table 2 ijerph-18-06768-t002:** The relationship between baseline measures of hearing thresholds at various frequencies and anthropometric variables, comorbid diseases, hearing-specific history, and severities of hearing impairment and sleep-disordered breathing by univariate analysis.

			Low Frequency, dB				Medium Frequency, dB				High Frequency, dB				Average Frequency, dB			
	N =	(%)	Mean ± SD; *r*			*p*	Mean ± SD; *r*			*p*	Mean ± SD; *r*			*p*	Mean ± SD; *r*			*p*
Gender						*				*				***				**
Male	42	(55)	37.3	±	22.1		37.7	±	26.2		63.0	±	25.2		46.8	±	24.2	
Female	35	(45)	27.7	±	14.7		25.9	±	14.7		44.1	±	19.6		32.2	±	13.7	
Smoking						**				**				***				***
No	57	(74)	28.3	±	14.1		26.0	±	15.6		46.9	±	21.6		33.8	±	16.0	
Yes	20	(26)	46.0	±	26.4		50.3	±	28.8		75.9	±	20.0		58.3	±	24.2	
Alcohol						*				NS				**				*
No	50	(65)	29.1	±	16.1		28.7	±	18.2		48.8	±	22.3		35.7	±	17.4	
Yes	27	(35)	40.0	±	23.4		39.0	±	27.7		64.8	±	25.8		48.4	±	25.4	
Coronary artery disease						NS				*				**				*
No	59	(77)	31.7	±	19.2		28.8	±	21.4		50.4	±	24.0		36.9	±	20.4	
Yes	18	(23)	36.9	±	20.9		44.0	±	22.4		67.6	±	22.6		50.9	±	21.0	
Hypertension						NS				*				**				*
No	44	(57)	31.4	±	17.7		26.6	±	18.3		47.8	±	22.1		34.8	±	17.3	
Yes	33	(43)	35.0	±	21.9		40.1	±	25.2		63.3	±	25.4		47.4	±	24.1	
Diabetes mellitus						NS				NS				NS				NS
No	65	(84)	33.2	±	20.4		32.3	±	23.5		54.0	±	25.5		40.1	±	22.2	
Yes	12	(16)	31.3	±	14.9		32.8	±	16.2		56.7	±	20.0		40.5	±	15.9	
Hypercholesterolemia						NS				NS				NS				NS
No	63	(82)	32.1	±	20.2		31.5	±	23.2		53.6	±	25.3		39.5	±	22.1	
Yes	14	(18)	36.4	±	16.6		36.0	±	18.7		58.0	±	21.7		43.4	±	17.3	
Depression						NS				*				NS				NS
No	72	(94)	33.7	±	20.0		32.8	±	23.1		54.3	±	24.7		40.6	±	21.9	
Yes	5	(6)	22.0	±	5.7		25.3	±	4.0		55.5	±	26.1		34.8	±	7.8	
Sudden deafness						NS				NS				NS				NS
No	57	(74)	32.3	±	19.1		31.8	±	22.3		55.4	±	23.5		39.8	±	20.2	
Yes	20	(26)	34.8	±	21.1		33.8	±	23.2		51.8	±	28.1		41.3	±	24.6	
Duration of hearing impairment						NS				NS				NS				NS
<6 m	29	(38)	31.9	±	20.3		31.2	±	23.8		50.6	±	28.4		38.8	±	24.9	
6 m~3 y	18	(23)	28.3	±	10.4		27.4	±	13.0		48.6	±	18.9		34.6	±	12.3	
>3 y	30	(39)	36.7	±	22.6		36.4	±	25.2		61.6	±	22.7		44.8	±	21.4	
Age	77		0.18			NS	0.32			**	0.50			***	0.41			***
BMI	77		0.15			NS	0.16			NS	−0.01			NS	0.10			NS
AHI	77		0.26			*	0.43			***	0.28			*	0.41			***
miniSpO_2_	77		−0.10			NS	−0.17			NS	−0.22			NS	−0.18			NS

Abbreviations: dB, decibels; SD, standard deviation; 6 m, 6 months; 3 y, 3 years; BMI, body mass index; AHI, apnea–hypopnea index; miniSpO_2_, lowest oxyhemoglobin saturation; NS, non-significance. * *p* < 0.05, ** *p* < 0.01, *** *p* < 0.001.

**Table 3 ijerph-18-06768-t003:** The effect of CPAP therapy on hearing thresholds by a generalized estimating equation with adjusting potential confounding factors.

	N	Low Frequency, dB Mean ± SD; *p* D: Mean ± SEm CI	Medium Frequency, dB Mean ± SD; *p* D: Mean ± SEm CI	High Frequency, dB Mean ± SD; *p* D: Mean ± SEm CI	Average Frequency, dB Mean ± SD; *p* D: Mean ± SEm CI
CPAP-BL	28	33.6 ± 20.7	36.7 ± 24.1	57.4 ± 25.6	43.7 ± 22.9
CPAP-3 m	28	29.6 ± 17.2 *p =* 0.111 D: −3.21 ± 2.01 CI: −7.15~0.73	34.3 ± 20.8 *p =* 0.106 D: −3.19 ± 1.97 CI: −7.05~0.67	57.8 ± 23.8 *p =* 0.784 D: 0.63 ± 2.28 CI: −3.84~5.10	41.4 ± 19.5 *p =* 0.621 D: −1.06 ± 2.14 CI:−5.25~3.13
CPAP-6 m	27	28.0 ± 15.9 *p =* 0.016 *** D: −5.60 ± 2.33 CI: −10.12~−1.03	33.4 ± 19.7 *p =* 0.023 *** D: −5.82 ± 2.56 CI: −10.84~−0.80	58.1 ± 21.6 *p =* 0.535 D: −2.37 ± 2.55 CI: −7.37~2.63	40.6 ± 18.3 *p =* 0.024 *** D: −5.10 ± 2.26 CI: −9.53~−0.67
CPAP-12 m	24	25.6 ± 15.3 *p =* 0.004 **** D: −7.97 ± 2.74 CI: −13.34~−2.60	29.4 ± 17.5 *p* = 0.001 **** D: −8.15 ± 2.35 CI: −12.76~−3.54	55.1 ± 20.8 *p =* 0.332 D: −2.80 ± 2.885 CI: −8.45~2.85	37.4 ± 16.7 *p =* 0.005 **** D: −6.67 ± 2.37 CI: −11.32~−2.02

Abbreviations: CI, confidence intervals (95% confidence intervals of the difference); D: Difference (comparing to the baseline); db, decibels; SD, standard deviation; SEm, standard error of the mean; CPAP, continuous positive airway pressure therapy; BL, 3 m, 6 m, and 12 m, corresponding to hearing threshold data measured at baseline, three months, six months, and 12 months. * *p* < 0.05, ** *p* < 0.01.

## Data Availability

The data presented in this study are available on request from the corresponding author. The data are not publicly available because we will try to do a retrospective cohort study in the future.

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
