# Peer review of "CPAP Treatment Improves Pure Tone Audiometry Threshold in Sensorineural Hearing Loss Patients with Sleep-Disordered Breathing"

_ijerph, 2021, doi:10.3390/ijerph18136768_

Round 1
Reviewer 1 Report
This retrospective observational cohort on Sensorineural hearing loss (SNHL) and Sleep-disordered breathing (SDB) is an interesting work, showing how CPAP could improve hearing in such patients.
Therefore, I recommend its publication after several improvements are done:
Lines 173-189: Selection of CPAP versus conservative medical treatment was not randomized. Instead, those in CMT did not accept CPAP. Please analyze potential bias. In table 1, those on CMT were mild OSA (AHI 19 vs 42), and were more commonly female, non-smokers and non-drinkers… It seems biased…
Lines 287-295: Using. GEE to express results of hearing changes makes comparison difficult. Data are shown in in Figure 2, but “traditional” statistics are missing, such as paired McNemar or similar. Looking the figure, it is amazing an improvement from 35 dB to 25 dB after CPAP in low-median frequencies… I would like to see its significance and confidence intervals.
Lines 314-317: The authors hypothesize the reason for a hearing improvement after CPAP in low-median frequencies. They quote a paper from 1972. Being the core of this paper, please add some references on hearing impairment and SDB:
- Matsumura E, Matas CG, Sanches SGG, Magliaro FCL, Pedreño RM, Genta PR, Lorenzi-Filho G, Carvallo RMM. Severe obstructive sleep apnea is associated with cochlear function impairment. Sleep Breath. 2018 Mar;22(1):71-77. doi: 10.1007/s11325-017-1530-5
- İriz A, Düzlü M, Köktürk O, Kemaloğlu YK, Eravcı FC, Küükünal IS, Karamert R. The effect of obstructive sleep apnea syndrome on the central auditory system. Turk J Med Sci. 2018 Feb 23;48(1):5-9. doi: 10.3906/sag-1705-66.
- Li X, Chen WJ, Zhang XY, Liang SC, Guo ZP, Lu ML, Ye JY. Inner ear function in patients with obstructive sleep apnea. Sleep Breath. 2020 Mar;24(1):65-69. doi: 10.1007/s11325-019-01891-7.
- Lisan Q, van Sloten T, Climie RE, Boutouyrie P, Guibout C, Thomas F, Danchin N, Jouven X, Empana JP. Sleep apnoea is associated with hearing impairment: The Paris prospective study 3. Clin Otolaryngol. 2020 Sep;45(5):681-686. doi: 10.1111/coa.13557.
- Gozeler MS, Sengoz F. Auditory Function of Patients with Obstructive Sleep Apnea Syndrome: A Study. Eurasian J Med. 2020 Jun;52(2):176-179. doi: 10.5152/eurasianjmed.2019.18373.
- Deniz M, Ersözlü T. Evaluation of the changes in the hearing system over the years among patients with OSAS using a CPAP device. Cranio. 2020 Jun 28:1-4. doi: 10.1080/08869634.2020.1783050.
Author Response
To the reviewer:
June 8, 2021
Dear Reviewer,
We have revised our draft and resubmitted for your consideration the article, “Sleep-Disordered Breathing with CPAP Treatment Improves Hearing Ability in Sensorineural Hearing Loss Patients “for inclusion in the International Journal of Environmental Research and Public Health.
Thank you for your important suggestions and we have already modified the draft.
- Concerning the most important part~ (Editor, Reviewer 1, and Reviewer 2)
“As baseline hearing parameters of the 2 groups are not similar it is not possible to compare the two treatments. Therefore we propose to remove the CMT group and only leave the CPAP group and show the effect of CPAP treatment on hearing parameters following the chosen time frames.”
- We discussed the relationship between covariate factors and hearing thresholds in all 77 participants. And because the significant difference in demographic measures between CPAP and non-CPAP groups could lead to a bias, when we discussed the effect of CPAP in SNHL participants with SDB, the case series study was chosen for data processing (n=28).
- Concerning Figure 2 and Table 3~ (Editor, Reviewer 1, and Reviewer 2)
“Data are shown in Figure 2, but “traditional” statistics are missing, such as paired McNemar or similar. Looking at the figure, it is amazing an improvement in low-median frequencies… I would like to see its significance and confidence intervals.”
“Figure 2 shows clearly why the two groups cannot be compared”
- We remade Figure 2 and Table 3 with traditional statistics and we also showed the significance and 95% CI. However, for adjusting confounding factors of hearing thresholds which we noted in Table 2, we finally chose GEE instead of paired T-test in Table 3 and Figure 2.
- For references~(Reviewer 1)
“The authors hypothesize the reason for a hearing improvement after CPAP in low-median frequencies. They quote a paper from 1972. Being the core of this paper, please add some references on hearing impairment and SDB”
- We rewrote our discussion with the current references. Thank you for your kind reminder.
- For hearing ability and thresholds~ (Reviewer 2)
“I don’t think that hearing ability should be used in the title and throughout the paper. The hearing ability not exactly threshold changes for pure-tone audiometry. The authors could consider improvement in pure-tone audiometry instead.”
- Thank you for your kind reminder. We renamed our title and used hearing / pure tone audiometry threshold instead of hearing ability in our draft.
All the authors have signed the letter to indicate that they have read and approved the paper. As principal investigator and the first author (Jessie Chao-Yun Chi), I take full responsibility for the integrity of the content of the article.
We trust that the enclosed submission meets the standards of your journal. However, if they do not, please do not hesitate to contact me should you need any additional information.
Thank you for your professional advice and wish you have a nice day~
Regards,
Jessie Chao-Yun Chi, MD; Shin-Da Lee, Ph.D.; Ren-Jing Huang, Ph.D.; Ching-Hsiang Lai, Ph.D.; Stanley Yung Liu, MD, DDS; Yih-Jeng Tsai, MD, Ph.D.; Po-Han Fu, Ph.D.; Hua Ting, MD

Reviewer 2 Report
Sleep-Disordered Breathing with CPAP Treatment Improves Hearing Ability in Sensorineural Hearing Loss Patients by Jessie Chao-Yun Chi, Shin-Da Lee, Ren-Jing Huang, Ching-Hsiang Lai, Stanley Yung Liu, Yih-Jeng Tsai, Po-Han, and Hua Ting
This is a retrospective observational study comparing in 77 human participants with sensory neuronal hearing loss and sleep-disordered-breathing. The aim of the study was to explore the effect on the hearing loss of the treatment the sleep apnoea with the use of the CPAP. After comparing two groups of patients treated with or without CPAP (n=28 and 49, respectively), the authors conclude that individuals treated with it presented an improvement after 6 and 12 months with better thresholds for low and medium frequencies.
The main problem is that the groups are not equally distributed with more profound hearing loss in the CPAP group. Unless both groups are comparable in hearing loss, I don’t see how the conclusion can be maintained. Figure 2 shows clearly why the two groups cannot be compared. The data shows that the CPAP group has worst thresholds than the control (CMT groups in the average frequency, abd also in the high and middle frequency. In fact, after 12 months with the CPAP, the pure-tone audiometry data in both groups are quite similar. Unless the authors are able to pull out another cohort the participants with similar audiometric basal data, I don’t see how comparisons can be done and conclusions reached. In addition, I don’t think that hearing abilities should be used in the title and throughout the paper. Hearing ability not exactly threshold changes for pure-tone audiometry. The authors could consider improvement in pure-tone audiometry instead.
Author Response
To the reviewer:
June 8, 2021
Dear Reviewer,
We have revised our draft and resubmitted for your consideration the article, “Sleep-Disordered Breathing with CPAP Treatment Improves Hearing Ability in Sensorineural Hearing Loss Patients “for inclusion in the International Journal of Environmental Research and Public Health.
Thank you for your important suggestions and we have already modified the draft.
- Concerning the most important part~ (Editor, Reviewer 1, and Reviewer 2)
“As baseline hearing parameters of the 2 groups are not similar it is not possible to compare the two treatments. Therefore we propose to remove the CMT group and only leave the CPAP group and show the effect of CPAP treatment on hearing parameters following the chosen time frames.”
- We discussed the relationship between covariate factors and hearing thresholds in all 77 participants. And because the significant difference in demographic measures between CPAP and non-CPAP groups could lead to a bias, when we discussed the effect of CPAP in SNHL participants with SDB, the case series study was chosen for data processing (n=28).
- Concerning Figure 2 and Table 3~ (Editor, Reviewer 1, and Reviewer 2)
“Data are shown in Figure 2, but “traditional” statistics are missing. Looking at the figure, it is amazing an improvement in low-median frequencies… I would like to see its significance and confidence intervals.”
“Figure 2 shows clearly why the two groups cannot be compared”
- We remade Figure 2 and Table 3 with traditional statistics and we also showed the significance and 95% CI. However, for adjusting confounding factors of hearing thresholds which we noted in Table 2, we finally chose GEE instead of paired T-test in Table 3 and Figure 2.
- For hearing ability and thresholds~ (Reviewer 2)
“I don’t think that hearing ability should be used in the title and throughout the paper. The hearing ability not exactly threshold changes for pure-tone audiometry. The authors could consider improvement in pure-tone audiometry instead.”
- Thank you for your kind reminder. We renamed our title and used hearing / pure tone audiometry threshold instead of hearing ability in our draft.
- For references~(Reviewer 1)
“The authors hypothesize the reason for a hearing improvement after CPAP in low-median frequencies. They quote a paper from 1972. Being the core of this paper, please add some references on hearing impairment and SDB”
- We rewrote our discussion with the current references. Thank you for your kind reminder.
All the authors have signed the letter to indicate that they have read and approved the paper. As principal investigator and the first author (Jessie Chao-Yun Chi), I take full responsibility for the integrity of the content of the article.
We trust that the enclosed submission meets the standards of your journal. However, if they do not, please do not hesitate to contact me should you need any additional information.
Thank you for your professional advice and wish you have a nice day~
Regards,
Jessie Chao-Yun Chi, MD; Shin-Da Lee, Ph.D.; Ren-Jing Huang, Ph.D.; Ching-Hsiang Lai, Ph.D.; Stanley Yung Liu, MD, DDS; Yih-Jeng Tsai, MD, Ph.D.; Po-Han Fu, Ph.D.; Hua Ting, MD
